# Keeping Myeloma in Check: The Past, Present and Future of Immunotherapy in Multiple Myeloma

**DOI:** 10.3390/cancers13194787

**Published:** 2021-09-24

**Authors:** James Ackley, Miguel Armenta Ochoa, Delta Ghoshal, Krishnendu Roy, Sagar Lonial, Lawrence H. Boise

**Affiliations:** 1Department of Hematology and Medical Oncology, Emory University, Atlanta, GA 30322, USA; james.ackley@emory.edu (J.A.); sloni01@emory.edu (S.L.); 2The Wallace H. Coulter Department of Biomedical Engineering, Georgia Institute of Technology and Emory University, Atlanta, GA 30332, USA; miguel.armochoa@gatech.edu (M.A.O.); delta@gatech.edu (D.G.); krish.roy@gatech.edu (K.R.); 3NSF Engineering Research Center for Cell Manufacturing Technologies, The Marcus Center for Therapeutic Cell Characterization and Manufacturing and the Center for ImmunoEngineering, The Parker H. Petit Institute for Bioengineering & Bioscience, Georgia Institute of Technology, Atlanta, GA 30332, USA; 4Winship Cancer Institute, Emory University, Atlanta, GA 30322, USA

**Keywords:** multiple myeloma, immunotherapy, IMiD, antibody, ADC, bi-specific antibody, CAR-T, vaccine

## Abstract

**Simple Summary:**

Multiple myeloma is the second most common hematological malignancy and while patients can have long responses to therapy, most patients will eventually develop treatment-resistant disease. Over the last thirty years, improved understanding of multiple myeloma and therapeutic advancements have dramatically improved outcomes for patients. Recently, advances in immunotherapy have revolutionized standard-of-care therapies and provided therapeutic options for patients with heavily pretreated, relapsed refractory multiple myeloma. Immunotherapy is a rapidly evolving field, and this review encompasses the immunotherapies that are currently used to treat myeloma patients as well as recent advances that are poised to advance myeloma therapeutics in the coming years.

**Abstract:**

Multiple myeloma is an incurable disease of malignant plasma cells and an ideal target for modern immune therapy. The unique plasma cell biology maintained in multiple myeloma, coupled with its hematological nature and unique bone marrow microenvironment, provide an opportunity to design specifically targeted immunotherapies that selectively kill transformed cells with limited on-target off-tumor effects. Broadly defined, immune therapy is the utilization of the immune system and immune agents to treat a disease. In the context of multiple myeloma, immune therapy can be subdivided into four main categories: immune modulatory imide drugs, targeted antibodies, adoptive cell transfer therapies, and vaccines. In recent years, advances in all four of these categories have led to improved therapies with enhanced antitumor activity and specificity. In IMiDs, modified chemical structures have been developed that improve drug potency while reducing dose limiting side effects. Targeted antibody therapies have resulted from the development of new selectively expressed targets as well as the development of antibody drug conjugates and bispecific antibodies. Adoptive cell therapies, particularly CAR-T therapies, have been enhanced through improvements in the manufacturing process, as well as through the development of CAR constructs that enhance CAR-T activation and provide protection from a suppressive immune microenvironment. This review will first cover in-class breakthrough therapies for each of these categories, as well as therapies currently utilized in the clinic. Additionally, this review will explore up and coming therapeutics in the preclinical and clinical trial stage.

## 1. Introduction

Multiple myeloma is a disease of malignant plasma cells and the second most common hematological cancer. As of 2020, myeloma accounts for 1.8 percent of new cancer diagnoses and for 2.1 percent of all cancer deaths [1]. In the last thirty years, exciting progress has been made in the treatment of myeloma, with five-year survival increasing from 31.2 percent to 53.9 percent [1]. This shift in prognosis was first driven by the use of autologous stem cell transplants, and then by the development of novel therapies such as proteasome inhibitors and immune modulatory imide drugs (IMiDs) that exploit the innate plasma cell biology of myeloma cells to induce selective target cell death [2,3]. While these steps have had a significant impact, a tremendous unmet need exists, as multiple myeloma patients develop refractory disease [4]. The next generation of breakthrough therapies and resulting increases in patient responses are likely to result from progress made in the development of immunotherapies to treat transplant-eligible and -ineligible patients, as well as relapsed refractory multiple myeloma.

Several characteristics of multiple myeloma form its therapeutic window in the context of immunotherapy and make it an attractive target. These include the selective expression of cell surface markers and manageable on-target off-tumor effects in the form of depleted humoral immunity resulting from plasma cell aplasia [5,6,7,8,9]. When combined, these factors allow the immune system to access the tumor, recognize tumor-associated antigens, and induce target cell death while avoiding dose-limiting toxicities. The breadth and rapid development of immunotherapies in multiple myeloma over the past decade necessitate an updated look at the therapies being developed and tested in clinical trials at present. Immunotherapy in multiple myeloma can be broadly divided into four categories: IMiDs, targeted antibodies, adoptive cell therapies, and tumor vaccines. In this review, we will discuss the history, current standard of care, and ongoing clinical prospects for each of these classes of drug.

## 2. IMIDs

IMiDs are one of the first successful immune-based therapies utilized in multiple myeloma, with thalidomide and its derivative lenalidomide gaining FDA approval in 2006 [10]. These drugs have become an essential aspect of the standard-of-care treatment regimen, with newly diagnosed patients typically receiving a combination of lenalidomide, bortezomib, and dexamethasone as an induction therapy in autologous stem-cell transplant (ASCT)-eligible patients and as the standard of care in transplant-ineligible patients [11,12]. Patients with relapsed refractory multiple myeloma often receive an IMiD as part of the new lines of therapy, as patients often remain sensitive to IMiDs upon relapse [13].

### 2.1. Mechanism of Action

The mechanism of action for IMiDs in myeloma has long been a debated topic due to its wide range of potentially anti-neoplastic effects. Thalidomide was initially of interest due to its anti-angiogenic properties; however, this has come into question with the discovery of additional anti-neoplastic properties and the success of lenalidomide and pomalidomide, which have reduced anti-angiogenic properties compared to thalidomide [14]. IMiDs have demonstrated direct cytotoxic effects, an influence over cytotoxic lymphocytes, and the ability to disrupt myeloma bone marrow stromal cell (BMSC) interaction [10].

Recent discoveries have attributed the anti-myeloma effects and enhanced T-cell activation to the binding of IMiDs to cereblon (CRBN). CRBN complexes with cullin 4A (CUL4A) or cullin 4B (CUL4B), which act as scaffolds to coordinate association with the adaptor protein DNA damage binding protein 1 (DDB1), and ring finger protein Roc1 to form a Cullin-RING E3 ubiquitin ligase (CRL), in this case, CRL4 E3 ligase [15,16]. CRL4 controls the degradation of transcription factors Ikaros (IKZF1) and Aiolos (IKZF3) [10]. The binding of IMiDs to CRBN increases its affinity for Ikaros and Aiolos, promoting their degradation [17].

In addition to their direct cytotoxic effects, IMiDs also enhance the activity of cytotoxic T cells and NK-cells through a variety of mechanisms. IMiDs enhance cytotoxic T cell activity by promoting T cell expansion and reducing the need for co-stimulation as Ikaros acts as an inhibitory regulator of the IL2 promotor [18,19,20]. Removing the inhibitory effect of Ikaros allows the T cell to induce target cell death without CD28-based co-stimulation and can even partially override cytotoxic T-lymphocyte-associated protein 4 (CTLA4) induced inhibitory signaling [19]. Increased levels of IL2 enhance T cell proliferation and natural killer cell (NK cell) activity [20]. Treatment with IMiDs can further augment the immune response by increasing antigen presentation in bone-marrow-associated dendritic cells, increasing the ratio of effector T cells to T-regulatory cells, and inhibiting the production of inflammatory cytokines [14,21,22,23].

Lastly, IMiDs have been demonstrated to disrupt the myeloma bone marrow microenvironment relationship in several ways. Treatment with IMiDs decreases the expression of vascular-cell-adhesion molecule 1 (VCAM-1) and intercellular adhesion molecule 1 (ICAM-1), which play key roles in the adhesion of multiple myeloma to the bone marrow compartment [24]. Further, IMiDs decrease tumor-supportive BMSC cytokine signaling in the form of reduced vascular endothelial growth factor (VEGF), IL6, tumor necrosis factor alpha (TNF-a), and transforming growth factor beta (TGFB) [14,24].

### 2.2. History

Developed in Germany during the early 1950s, thalidomide was often prescribed as a sedative to treat insomnia, and morning sickness in pregnant women. It is chemically similar to barbiturates and was seen as a safe alternative [25]. The usage of thalidomide for this indication was common across the world, before it was pulled from the market in the early 1960s due to reports of birth defects in the children of women who took thalidomide while pregnant [25]. Thalidomide was never approved for use in the United States due to concerns about the drug’s safety regarding the induction of neuropathy and lack of a sufficient safety profile raised by Frances Kelsey [25,26].

There are three IMiDs currently approved by the FDA for use in multiple myeloma: thalidomide, lenalidomide, and pomalidomide [10]. Although they are structurally similar, modest structural changes have resulted in dramatic shifts in potency and drug metabolism [10].

### 2.3. Thalidomide

Thalidomide is a first-in-class IMiD and is still used to treat newly diagnosed and relapsed refractory multiple myeloma (RRMM) [13,27]. It is the least potent of the approved IMiDs and, in the United States, is typically reserved for the treatment of patients who no longer respond to lenalidomide [13]. This is partially due to adverse events resulting in low tolerability. Thalidomide-related adverse events include somnolence, constipation, fatigue/weakness, dizziness, tremors and potentially permanent nerve damage and peripheral neuropathy [10,13,28]. Although it was being examined as an antineoplastic agent as early as the 1960s, it was not until the discovery of its antiangiogenic properties by R J D’ Amato in the 1990s that interest in the potential of thalidomide as an antineoplastic agent [29,30]. Initial clinical trials examining the use of thalidomide in multiple myeloma revealed that 29% of RRMM patients experienced a partial response or better when receiving thalidomide as a single agent [10,31]. Further, when given a combination of thalidomide, dexamethasone, and a chemotherapeutic agent, this number rose to approximately three quarters of patients experiencing a partial response or better [31]. Compared to lenalidomide, thalidomide is less myelosuppressive, making it a useful tool in cases where the patient cannot receive blood-product support, such as Jehovah’s witnesses [32,33].

### 2.4. Lenalidomide

Lenalidomide, in combination with bortezomib and dexamethasone, is an essential part of the current standard of care for newly diagnosed multiple myeloma (NDMM) patients [10]. Approved in the same year as thalidomide, lenalidomide is the preferred choice due to its improved progression-free survival and tolerability [34,35]. In its debut clinical trial, lenalidomide proved to be effective and well-tolerated even in patients who had previously been treated with thalidomide [35]. As a single agent, lenalidomide was effective in 47% of newly diagnosed multiple myeloma patients. Dose-limiting toxicities were primarily hematological in nature and consisted of neutropenia, thrombocytopenia, and anemia. In a clinical trial, NCT00378105, when given to newly diagnosed multiple myeloma patients in combination with bortezomib and dexamethasone, lenalidomide was observed to lead to a partial response rate of 100%, with the majority having a very good partial response [36]. When examining this drug combination in cases of relapse and refractory multiple myeloma, 64% of patients, many of whom had previously underwent IMiD-based therapy, experienced a partial response or better.

### 2.5. Pomalidomide

Most recently in 2013, pomalidomide was approved for RRMM patients who have undergone at least two prior lines of therapy consisting of bortezomib and lenalidomide [10]. In a phase 2 clinical trial, pomalidomide demonstrated an overall response rate of 63%, including a 40% response rate in patients refractory to lenalidomide and a 37% response rate in patients’ refractory to thalidomide [37]. Although structurally similar to thalidomide and lenalidomide, pomalidomide is primarily metabolized in the liver by CYP1A2 and CYP34A. This prevents pomalidomide from accumulating in patients with renal insufficiency, as seen with lenalidomide, but increases the likelihood of drug–drug interactions [10,38].

### 2.6. Future Direction

#### 2.6.1. Iberdomide

Iberdomide is a recently developed, highly potent IMiD that is being evaluated for use in both NDMM and RRMM [39]. Preclinical studies in myeloma cell lines have demonstrated more potent antiproliferative effects, and increased degradation of Ikaros and Aiolos compared to pomalidomide [39]. The 1b/2a clinical trial NCT02773030 examines iberdomide as a single agent, in combination with dexamethasone, and as part of the triple regimen of iberdomide, dexamethasone, and daratumumab/bortezomib [40]. To date, preliminary results have been published for the combination of iberdomide and dexamethasone in RRMM. In this cohort, 100% of patients were exposed to lenalidomide and 69% of patients were exposed to pomalidomide. The results have been promising, with 31% of patients achieving a partial response or better, 51% of patients achieving a minimal response or better, and 88% of patients achieving stable disease or better [40]. Additionally, its favorable toxicity profile and efficacy in the setting of pomalidomide-resistant disease suggests that iberdomide may be a useful tool in treating multiple myeloma in the future [40].

#### 2.6.2. IMIDs in SMM

Multiple myeloma is a disease of malignant plasma cells and preceded by two asymptomatic conditions, monoclonal gammopathy of undetermined significance (MGUS) and smoldering multiple myeloma (SMM). MGUS is characterized as less than 3 g/dL of serum M-protein and under 10% of a clonal plasma cell population. SMM is characterized as more than 3 g/dL of serum M-protein and a clonal population of plasma cells between 10% and 60% [41]. Both of these conditions are additionally defined by the lack of a myeloma-defining event including hypercalcemia, renal failure, anemia and bone lesions [42]. As of 2018, the Mayo clinic and the International Myeloma Working Group describe three attributes that define high-risk SMM. These include serum monoclonal protein levels greater than 2 g/dL, an involved-to-uninvolved, serum-free, light-chain ratio greater than 20, and greater than a 20% clonal bone-marrow plasma-cell population. Low-risk, intermediate-risk, and high-risk SMM are defined as having 0, 1, or 2–3 of these attributes and represent a two-year rate of progression of 5%, 17%, and 46%, respectively [43,44]. Additionally, cytogenetic abnormalities such as t(4;14), t(14;16), +1q, and/or del13q can be used to further divide SMM into four categories: low risk, low intermediate risk, intermediate risk, and high risk, with 6%, 23%, 46%, and 63% risk of two-year progression, respectively [44]. The current standard of care for the myeloma precursor condition SMM is observation; however, recent data are challenging this belief [43]. Two strategies for managing SMM are currently being investigated in clinical trials. These include a preventative approach, with the goal of preventing the progression of SMM to multiple myeloma, and a more intensive curative approach, with the goal of eradicating the premalignant state [42]. This preventative approach uses IMiD therapy as a single agent or in combination with dexamethasone and or Datatumumab. The QuiRedex study examined the use of Lenalidomide and dexamethasone in high-risk SMM. Progression-free survival in the Len + Dex treatment group was not reached as of the last update released for the study, compared to a median progression-free survival of 23 months in the observation group [45]. The median follow-up time for surviving patients was 75 months [45]. While median overall survival was not reached, at the time of data cut-off, 18% of patients died in the treatment group compared to 36% in the control group [45]. This was followed by the E3A06 trial, which examined the use of Lenalidomide as a single agent in preventing the progression of high-risk SMM to MM. Patients who received Lenalidomide demonstrated 1, 2, and 3 year progression-free survival rates of 98%, 93%, and 91%, compared to 89%, 76%, and 66% in the observation control group [46].

## 3. Antibody Based Therapies

The use of therapeutic antibodies to treat cancer has been part of the immunotherapy boom of the past two decades. Multiple myeloma is no exception, with a number of effective antibody-based therapies having been approved in recent years [47,48,49,50]. Antibody-based therapies in multiple myeloma generally fall into one of three general categories: monoclonal antibodies, antibody–drug conjugates (ADCs), and bi-specific antibodies. These therapies utilize the highly specific antigen recognition abilities of antibodies to induce selective target-cell death. How they accomplish this varies.

### 3.1. Monoclonal Antibodies

The last decade has seen several successes in the development of therapeutic antibodies in multiple myeloma. Plasma cells are incredibly unique cells and myeloma cells often overexpress cell-surface proteins exclusive to their plasma cell and B cell lineage. This results in selective antigens that can be targeted by monoclonal antibodies to induce selective target cell death. At present, three monoclonal based antibody therapies have been approved for use in RRMM in the form of first-in-class elotuzumab (2015), daratumumab (2015), and, more recently, isatuximab (2020) [47,48,49]. It is important to note that, at therapeutically relevant concentrations, these therapeutic antibodies can generate a false positive result in an SPEP test measuring M-protein levels. This can be controlled for though the use of mass spectrometry as opposed to an SPEP test [51]. Although currently approved for use in RRMM, there is an active effort to incorporate monoclonal antibody therapy into the conditioning regimens and as a front-line therapy in newly diagnosed multiple myeloma (NDMM) (NCT02252172).

#### 3.1.1. Mechanism of Action

Monoclonal antibodies induce target cell death through three main mechanisms: antibody-dependent cellular cytotoxicity (ADCC), antibody-dependent cellular phagocytosis (ADCP), and compliment-dependent cytotoxicity (CDC), as shown in Figure 1A [52]. ADCC is driven by NK cells, which engage the FC region of the antibody through FC receptors such as CD16 to trigger the degranulation of perforin and granzymes or the engagement of CD95 to induce cellular apoptosis [53]. NK cells are well-suited to engage in antitumor immunity, as NK cell activation is inhibited by interaction with MHC class I molecules, which are frequently downregulated in transformed cells [54]. CD16 is likewise expressed on phagocytic macrophages, along with CD32 and CD64, which also bind to the FC region of therapeutic antibodies to induce target-cell phagocytosis [55]. Although sometimes overlooked, it has been demonstrated in murine models that ADCP has a comparable impact on reducing the tumor burden to NK cell-targeted killing [55]. Finally, the FC region of monoclonal antibodies can induce a complement cascade, which can result in the direct lysis of target cells or the further opsonization to promote ADCP [56,57].

#### 3.1.2. History

The first attempt to use a monoclonal antibody in multiple myeloma came in the form of rituximab. Rituximab is a chimeric human/mouse antibody targeting CD20 [58]. Rituximab has successfully been used to treat a number of B-cell, non-Hodgkin lymphomas and is part of the standard of care in some cases, along with chemotherapy regimens [58]. This success, along with the identification of CD20+ myeloma cells in a subset of patients, encouraged the idea of evaluating the efficacy of rituximab in multiple myeloma [58,59]. However, the results were disappointing, with rituximab providing no benefit to patients with RRMM [59].

#### 3.1.3. Elotuzumab

Elotuzumab is a humanized monoclonal antibody targeting the glycoprotein signaling lymphocyte activation molecule F7 (SLAMF7), also known as cell-surface glycoprotein CD2 subset 1 (CS1) (Figure 1B) [60]. SLAMF7 is highly expressed on multiple myeloma cells and more modestly expressed on NK cells and a subset of T cells [61]. In addition to the standard mechanisms of action attributed to monoclonal-based antibody therapy, elotuzumab is known to enhance NK cell activity through the engagement of SLAMF7. Crosslinking elotuzumab with SLAMF7 recruits the adaptor protein EAT-2, which activates PLCy, resulting in calcium flux and an increase in NK cell activity [54]. This activating signaling does not occur in myeloma cells due to a lack of EAT-2 expression.

Elotuzumab received FDA approval for use in RRMM in 2015 in response to the ELOQUENT-2 (NCT01239797) phase III clinal trial, which evaluated the addition of elotuzumab to a regimen of lenalidomide and dexamethasone [47]. Elotuzumab, in combination with lenalidomide and dexamethasone (ELd), improved both progression-free survival and overall survival compared to the control group receiving lenalidomide and dexamethasone (Ld), without an increase in adverse events [47]. Progression-free survival in the ELd arm of the study at one and two years was 68% and 41%, respectively, compared to 57% and 27% in the control group [47]. Overall survival at one, two, and three years improved from 83% to 91%, 69% to 73%, and 53% to 60%, respectively. Final overall survival numbers were reported in 2020, revealing an increase of 8.7 months in patients who received ELd as opposed to Ld [62]. Since then, elotuzumab has become a staple in the treatment of RRMM and is often used in combination with IMiDs and proteosome inhibitors [63].

More recently, in ELOQUENT-3 (NCT02654132), elotuzumab was examined in combination with pomalidomide and dexamethasone in patients who were refractory or were relapsed refractory to lenalidomide and a proteosome inhibitor. The addition of elotuzumab to pomalidomide and dexamethasone increased response rates from 26% to 53% and progression-free survival from 4.7 to 10.3 months [64]. At the 12-month and 18-month timepoints, progression-free survival in the elotuzumab group was 43% and 34%, respectively, compared to 20% and 11% in the control group [65]. Both the elotuzumab-containing group and the control group demonstrated similar toxicity profiles, with 57% of patients in the elotuzumab group experiencing a grade 3 or 4 adverse event compared to 60% in the control group [64].

#### 3.1.4. Daratumumab

Daratumumab is a human monoclonal antibody targeting CD38 (Figure 1B). CD38 is highly and consistently expressed on myeloma cells and, to a lesser degree, other hematological and non-hematological tissues [49]. CD38 plays a dual role in myeloma cell biology. It can act as an ectoenzyme converting NAD^+^ to adenosine to promote an immunosuppressive environment or as a receptor ligation, which promotes proliferation [66,67]. As was the case with elotuzumab, daratumumab’s anti-tumor activity is augmented through a non-standard mechanism of action for monoclonal antibodies. Examination of patient peripheral blood before, during, and after treatment with daratumumab revealed the depletion of a highly immunosuppressive CD38^+^ T-reg cell population [68]. This depletion was associated with an increase in cytotoxic, helper, and memory T cells [68].

Daratumumab received accelerated approval for use in RRMM in 2015 after two clinical trials (NCT01985126, NCT00574288), and demonstrated its potent anti-cancer activity as a single agent at 16 mg/kg in heavily pretreated patients. In NCT01985126, also known as the SIRIUS study, a response rate of 29.2% was observed including three CRs, ten VGPRs, and 18 PRs. The median duration or response was 7.4 months, and 12-month survival was 64.8% [49]. NCT00574288, also known as Gen501, showed a response rate of 36%, comprised of 2 CRs, 2 VGPRs, and 11 PRs. The median duration of response was 5.6 months and, of the patients who responded, 65% were progression-free at 12-months [69]. In combination, these studies comprise 146 patients at a 16 mg/kg dose, 86.5% of which were refractory to proteosome inhibitors as well as IMiDs [70]. The development of daratumumab marked a major milestone in the development of immune therapies for multiple myeloma.

Since its approval, daratumumab has been evaluated for use in combination with therapy with IMiDs and proteosome inhibitors. In the CASTOR study, adding daratumumab to a combination of bortezomib and dexamethasone improved 12-month progression free survival from 26.9% to 60.7% [71]. Median progression-free survival in the control group (Bd) was 7.2 months, and 16.7 months in the daratumumab-treated group (DBd). When combined with lenalidomide in the POLLUX study, daratumumab improved progression-free survival at the 13.5-month interim analysis point, from 41% in the control group (Ld) to 18.5% in the daratumumab-treated group (DLd) [72].

Daratumumab is also being evaluated in combination with lenalidomide, bortezomib, and dexamethasone for use in induction, consolidation, and maintenance therapy in transplant-eligible NDMM, as part of the GRIFFIN trial (NCT02874742). A total of 42.2% of patients who received daratumumab achieved an sCR, compared to 32% in the control arm. Daratumumab also increased the overall response rate from 91.8% to 99% [73]. Responses continued to deepen over time, with additional maintenance therapy resulting in an sCR rate of 63.3% in the daratumumab arm compared to 47.4% in the control arm [74].

Further, daratumumab has been used as part of the front-line treatment in transplant-ineligible patients, in combination with chemotherapies and anti-inflammatory agents.

#### 3.1.5. Isatuximab

Approved in March 2020, isatuximab is like daratumumab in that it is a humanized antibody targeting CD38 (Figure 1B) [75]. However, it does target a different amino acid sequence and is more potent, with an overall response rate as a single agent of 32% when dosed at 10 mg/kg QW, compared to a 31% combined ORR with daratumumab when dosed at 16 mg/kg QW. Interestingly, while daratumumab has been shown to induce the release of CD38 from the surface of myeloma cells, this was not seen with isatuximab [76].

Isatuximab is being examined in combination with pomalidomide and dexamethasone in the ICARIA study (NCT02990338). With isatuximab added to a regimen of pomalidomide, progression-free survival was extended from 6.5 to 11.5 months [48].

The IKEMA study (NCT03275285) is examining isatuximab in combination with carfilzomib in patients with relapsed or refractory multiple myeloma. Progression-free survival in the isatuximab group was not reached, while progression-free survival in the control group was 19.15 months [77].

Targeting CD38 with daratumumab and isatuximab was proven to be efficacious but can complicate the phenotyping of patient blood due to low levels of CD38 expression on red blood cells. CD38 bound to red blood cells causes panagglutination, resulting in a false-positive indirect antiglobulin test. While methods to circumvent this risk exist, it is crucial that this be accounted for, particularly in emergency situations [78].

### 3.2. Immune Checkpoint Inhibitors

Immune checkpoint inhibitors have demonstrated efficacy in a variety of tumor types and function by targeting immune-suppressive signaling in the tumor microenvironment [79]. This is commonly performed through monoclonal antibodies targeting the PD-1 PD-L1 interaction (Figure 1A), or CTLA-4, which binds to the immune stimulatory CD80 and CD86 to induce an immune-suppressive signal [79,80]. Despite the elevated levels of PD-L1 in multiple myeloma cells, clinical trials examining the use of pembrolizumab as a single agent failed to demonstrate its efficacy in multiple myeloma [81,82]. This is potentially due to the immune dysplasia induced by multiple myeloma, as well as the highly immune-suppressive bone-marrow microenvironment [83]. When investigated in combination with IMiD therapy and dexamethasone as part of the KEYNOTE-183, and KEYNOTE-185 trials, pembrolizumab appeared to enhance the anti-neoplastic effects of therapy but was associated with increased risk of adverse events [84,85]. This led to the trials being halted at the request of the FDA [84,85].

### 3.3. Antibody Drug Conjugates

Similar to monoclonal antibody-based therapies, antibody drug conjugates (ADCs) utilize the highly specific antigen recognition abilities of antibodies to induce selective target cell death. ADCs are composed of three distinct but equally important components, a monoclonal antibody, a linker, and a cytotoxic agent (Figure 1B) [86]. The antibody and the linker work in tandem act to open the therapeutic window of drugs that are too toxic to use in their unconjugated forms. At present, there is only one ADC therapy approved for use in multiple myeloma in the form of belantamab mafodotin, which targets BCMA (Figure 1B) [87]. This is not to say that this is not an active area of research, however, as ADCs targeting CD46, CD48, CD74, CD138, CD307, CD319, CD352, and ASCT2 have all reached clinical trials for treatment in multiple myeloma, with varying levels of success [88].

#### 3.3.1. Mechanism of Action

While the cytotoxic mechanism of action varies depending on the drug conjugate, the mechanism through which ADCs deliver their payload has been characterized. After intravenous dosing, ADCs circulate until they reach the bone marrow microenvironment, where they engage their target. At this point, the monoclonal antibody can engage ADCC, CDC, or ADCP to induce cell death in the same manner as monoclonal-based therapies (Figure 1A) [86]. Alternatively, an ADC binds to its target antigen and is endocytosed by the target cell, where the linker or monoclonal antibody is degraded, resulting in the release of the drug conjugate into the target cell [89]. The cytotoxic agent then takes effect, inducing target cell apoptosis. After the destruction of the target cell, it is possible for the cytotoxic agent to enter neighboring cells, where it can induce apoptosis in additional tumor cells or supportive stromal cells [86].

Selection of the proper cytotoxic agent is crucial for an effective ADC. While the use of antibodies and linkers can reduce the systemic effects of highly potent cytotoxic compounds, certain models estimate that less than 2% of the administered dose actually reaches the cytosol of target cells [90]. This necessitates the use of highly cytotoxic agents that are effective at low doses [86]. Common cytotoxic agents used in ADCs include microtubule agents such as dolastatin derivatives MMAE and MMAF, or maytansinoids DM1 and DM4, all of which are considered too toxic for use on their own [88,91,92]. Of note, MMAF- and maytansinoid-derived cytotoxic payloads have been associated with dose-limiting ocular toxicities in a variety of ADCs across a number of tumor types [93]. This ocular toxicity most frequently manifests as blurred vision, dry eyes, and corneal abnormalities [93].

Linkers are a critical component of the ADC and are key to the design of a safe, potent, and selective ADC [86]. Developments in linker technology have greatly increased the ability of ADCs to selectively deliver their payload to target cells while limiting their systemic effects [86]. Linkers generally fall into one of two categories: cleavable and non-cleavable linkers. Cleavable linkers rely on differences in the blood microenvironment and intracellular tumor environment to induce linker degradation. These take three forms: hydrazone, disulfide and peptide linkers, each of which is cleaved or degraded through a different mechanism [86]. Hydrazone linkers are pH-sensitive and undergo acidic hydrolysis in the lysosomal microenvironment [86]. Disulfide linkers are degraded by thiols, which commonly accumulate in tumor cells, where they act as antioxidants [86,94]. Peptide linkers are degraded by pH-dependent proteases, which are only active in the lysosome [86]. ADCs with non-cleavable linkers rely on the near-complete lysosomal degradation of the antibody to release the cytotoxic agent. This makes them entirely dependent on being endocytosed to release their drug product. However, this tradeoff provides enhanced stability while the ADC is in circulation, as well as improvements in the ADC half-life [86].

#### 3.3.2. History

The first ADC to enter clinical trials for use in multiple myeloma was BT062, which consisted of the maytansinoid DM4 conjugated to the chimeric anti-CD138 antibody nBT062 via a disulfide linker [95,96]. Phase 1 NCT00723359 and phase 1/2a NCT01001442 clinical trials examined the effect of BT062 as a single agent given as a single dose or multidose regimen, respectively [95]. The single-dose regimen resulted in a stable disease rate of 67.7% of patients, with a single patient experiencing a partial response resulting in an ORR of 3.2%, and two patients experiencing a minimal response [95]. A total of 77.4% of patients experienced stable disease or better. The multidose regimen faired similarly, with 61.8% of patients experiencing stable disease and an ORR of 5.9%, and three patients experiencing a minimal response. Cumulatively, 76.5% of patients on this regimen experienced stable disease or better [95]. These trials were followed up by examining the effects of BT062 in combination with IMiDs and dexamethasone in 2012; however, the results have not been reported (NCT01638936).

#### 3.3.3. Current Therapies

The only ADC currently approved for use in RRMM is belantamab mafodotin, which consists of the microtubule inhibitor MMAF conjugated to a humanized IgG1 antibody targeting BCMA by a non-cleavable maleimidocaproyl linker [87]. Accelerated first-in-class approval was granted to belantamab mafodotin in August 2020 as a result of the DREAMM-2 phase 2 clinical trial, which examined the effects of single-agent belantamab mafodotin in heavily pretreated patients refractory to both IMiDs and proteosome inhibitors, as well as patients refractory or intolerant to anti-CD38-based antibody therapy [50]. Belantamab mafodotin was given over a 21-day cycle at either 2.5 or 3.4 mg/kg. Response rates were 31% and 34%, of which 60% and 59% were very good partial responses or better, respectively. At the six-month follow-up point, median progression-free survival had not been reached, and the study is still ongoing.

This study was a follow-up to the two-part, phase I DREAMM-1 trial (NCT02064387). Part 1 of DREAMM-1 was used to determine the maximum tolerated dose (MTD), while part two acted as an expansion to evaluate the efficacy of the of the dose determined in part 1 (3.4 mg/kg) in RRMM. A total of 89% of patients in part 2 were double refractory to IMiDs and proteosome inhibitors, and 37% were refractory to the CD38 targeting Daratumumab. Median progression-free survival in this group was 7.9 months [97]. While room for improvement remains, belantamab mafodotin provides a therapeutic option for patients that have become refractory to standard-of-care treatments. 

The DREAMM-6 (NCT03544281) examines the use of belantamab mafodotin in combination with bortezomib and dexamethasone to treat relapsed refractory multiple myeloma. While the study is ongoing, the results are promising, with an initial ORR of 78% [98].

As is the case with many MMAF-containing ADCs, ocular toxicity is a dose-limiting adverse event in belantamab mafodotin treatment [99,100]. When dose-related adverse events cannot be mitigated through regular eye exams and the use of artificial tears, they are managed through dose reduction, dosing holidays, and discontinuation [99].

These DREAMM studies are part of a much larger investigational effort to characterize the role of belantamab mafodotin in multiple myeloma therapeutics. At this point in time, nine DREAMM trials have been completed, are active, or are recruiting, with a 10th trial currently being planned.

NCT03489525 examines MEDI2228, which targets BCMA and consists of a human antibody conjugated to a DNA cross-linking agent, pyrrolobenzodiazepine, through a protease-cleavable linker [101]. This phase 1 trial with an expansion arm established a maximum tolerated dose of 14 mg/kg, which was then used in an expansion cohort comprised of heavily pretreated patients who had progressed on standard-of-care agents including proteosome inhibitors, IMiDs, and CD38-targeted monoclonal antibodies [101]. This cohort saw an ORR of 61%, comprised of 10 VGPR and 15 PRs. The median duration of response has not been reached.

#### 3.3.4. Future

ADCs are a very promising and active avenue of drug development. At present, 13 clinical trials examining the effects of ADCs in multiple myeloma are in either the recruiting, enrollment, or active state according to a clinicaltrials.gov search for the term “Antibody Drug Conjugate” in multiple myeloma.

### 3.4. Bi-Specific Antibodies

Bi-specific antibodies are dual-targeting antibodies that can serve one of three main functions: immune effector cell redirection, tumor directed immune modulation, and dual immune modulation [102]. The most prominent bispecific antibodies engage tumor infiltrating immune effector cells by selectively binding to a tumor-associated antigen as well as a target on an immune effector cell (Figure 2). This serves to engage the immune effector cell towards malignant target cell and induce TCR-independent tumor cell death [103]. The most common effector and target antigens for bi-specific antibodies in multiple myeloma are CD3 and BCMA, respectively, with the majority of bi-specific antibodies evaluated in clinical trials consisting of this pairing. Additional tumor target antigens include GPRC5D, CD38, and FcRH5 (Table 1). All bi-specific antibodies currently being evaluated in clinical trials for multiple myeloma target CD3 on T cells, along with the tumor-associated antigen. There is a growing body of clinical and preclinical research examining the effects of targeting other immune target agents, including CD16A to engage NK cells, 4-1BB to engage activated tumor-infiltrating T cells, and dual-targeting immune checkpoint inhibitors [102].

Bi-specific antibodies take on a variety of forms and structures, with the key components being the antigen recognition domains and the linker. While the selection of antigen is crucial to induce the engagement of the bi-specific antibody and its target, the selection of a linker plays a major role in the pharmacokinetics and engagement of innate immune effector cells. Bispecific antibody linkers generally fall into one of two categories: scFV-based or full-length, IgG-like asymmetric antibodies [104]. scFV-based bi-specific antibodies include BiTEs, DARTs, and TandABs, all of which consist of scFV-binding regions connected by a short flexible linker. While these molecules have demonstrated potent anti-cancer activity, they are held back due to their short half-life, which results in the need for multi-week continuous IV dosing, and the associated production costs and challenges associated with this [104]. Recent work has been carried out to incorporate an FC region into scFV-based bi-specifics, which has been shown to increase the half-life of the bispecific in non-human primate models to a rate that would allow for weekly dosing [104]. Full-length IgG such as asymmetric antibodies are structured similarly to an IgG molecule and are generated through the combination of two hybridomas into a quadroma. Historically, the rate-limiting step in this process has been the correct assembly of the IgG molecule in the quadroma, with only 12.5% of antibodies forming the desired bi-specific. Advances in assembly technology, such as knobs-into-holes and the utilization of common heavy and common light chains, have made this process more efficient [104].

#### 3.4.1. Mechanism of Action

Immune effector redirecting bi-specific antibodies consist of two binding regions, one that targets an immune effector cell and one that targets a tumor-associated antigen. Bi-specific antibodies bind to their target antigen as well as tumor-infiltrating lymphocytes in the tumor microenvironment, creating a pseudo-immunological synapse. CD3 engagement on the TIL induces TCR-independent degranulation of perforin and granzymes to induce target cell death [105]. Additionally, Fc region containing bi-specific antibodies can engage the innate immune system and induce ADCC, CDC, and ADCP [106]. Tumor-directed, immune modulatory, bi-specific antibodies target a tumor-associated antigen as well as a T-cell costimulatory domain such as 4-1-BB to engage activated tumor-specific T cells with target cells [103]. The goal of this strategy is to enhance immunological memory by prioritizing the engagement of T cells that target a variety of tumor-associated antigens [103]. Dual immune modulatory bi-specific antibodies block the effects of the immunosuppressive microenvironment, often by targeting multiple immune checkpoints such as PDL-1 and LAG-3 [103].

The major adverse events associated with bi-specific antibody therapy include cytokine release syndrome (CRS) and neurotoxicity. CRS is described as an acute systemic inflammatory condition, driven by the release of cytokines by immune effector cells [107]. The presentation of CRS can vary but can include fever, fatigue, headache, hypoxia, altered mental state, and, in severe cases, hypotension [107,108]. CRS is typically treated through IL6R inhibition via tocilizumab. Corticosteroids are also used to treat severe CRS and can be used to treat neurological symptoms as tocilizumab does not cross the blood–brain barrier. Corticosteroids are typically not used as a front-line treatment for CRS as they induce a systemic immunosuppressive state that can impact the efficacy of immunotherapies [108,109]. Neurotoxicity presented in the context of CRS can include aphasia, headache, and cognitive disorder. These symptoms often resolve with successful CRS treatment [107].

#### 3.4.2. History

Bi-specific antibodies were first described in 1961 by Nisonoff and Rivers, who generated a bi-specific antibody through the pepsin digestion, and subsequent recombination of two monospecific antibodies [110]. Developments in hybridoma technology in 1975 led to the establishment of hybridomas or quadromas, which expressed two types of heavy and light chain antibody, resulting in the formation of bi-specific antibodies [111,112]. Evidence for the use of bi-specific antibodies as a cancer therapeutic came in 1985, when researchers demonstrated therapeutic activity in syngeneic and xenograph mouse models [113,114]. The first bi-specific antibody to gain regulatory approval was Catumaxomab, which targeted CD3 on T cells and EP-CAM on tumor cells and was approved in 2009 by the European Union for the treatment of malignant ascites in patients with EpCAM-positive disease [115]. This was followed in 2014 by the CD19 CD3 targeting Blinatumomab, which gained approval for use in relapse refractory B-ALL after a phase II clinical trial demonstrated complete or hematological remission in 43% of patients [116]. Blinatumomab was examined for use in multiple myeloma in clinical trial NCT03173430, but the study was terminated due to slow accrual [117].

#### 3.4.3. Future

The future of bi-specific antibody use in the treatment of multiple myeloma is brighter than ever, with an abundance of clinical and preclinical research driving the field forward.

AMG-420 and AMG-701 are a series of bi-specific antibodies targeting BCMA and CD3 [118]. AMG-420 was examined for use in RRMM as part of the phase 1 clinical trial NCT03836053. In this trial, heavily pretreated patients receiving 400 ug/d of MTFD experienced a 70% response rate consisting of five minimal residual disease-negative CRs, one VGPR, and one PR [118]. While these results are encouraging, AMG-420 faced challenges, including a short half-life, which necessitated the dose being given via extended continuous infusion [119]. While NCT03836053 remains active, it is no longer recruiting and the development of AMG-420 has been halted. To address the issues observed with AMG-420, AMG-701 was developed, which has an extended half-life due to its reduced renal clearance, facilitated by the addition of an FC region to the bispecific antibody. Preclinical studies of AMG-701 demonstrated its ability to induce T-cell-mediated toxicity towards multiple myeloma cell lines in immune-suppressive environments and T-cell-induced cell death in autologous patient myeloma cells [120]. AMG-701 is currently being examined in clinical trial NCT03287908. Patients receiving between three and 12 mg of AMG-701 experienced a response rate of 36%. Patients who received escalating doses of up to 9 mg demonstrated a response rate of 83%, consisting of three VGPR and three PRs.

CC-93269 is an asymmetric, two-arm, humanized antibody that utilizes the 2 + 1 format structure consisting of two affinity BCMA-binding domains with a lower affinity of CD3 binding. This structure serves to increase the avidity of the BCMA binding by utilizing two high-affinity BCMA-binding domains, along with a lower-affinity CD3-binding domain to decrease unspecific T cell activation [121,122]. CC-93269 is currently being examined for use in RRMM in phase 1 clinical trial NCT03486067. Patients who received more than 6mg in their first-cycle dose demonstrated a response rate of 83.3%, including seven very good partial responses or better, and four stringent complete responses (sCRs) [121].

PF-06863135 (Elrantamab) is currently being examined in three clinical trials, both as a single agent and in combination with IMiD therapy and dexamethasone. Of these three studies, preliminary results have been published for NCT03269136, which examines the effects of PF-06863135 as a single agent, subcutaneously, intravenously, or in combination with dexamethasone, lenalidomide, and pomalidomide. Reported results demonstrate an ORR of 33% at all dosing levels and 75% at the highest two dosing levels [123].

JNJ-64407564 (Talquetamab) is currently being examined in four clinical trials, both as a single agent and in combination with IMiD therapy. To date, results have been published for the phase 1 trial NCT03399799. While the goal of this study is to establish a safety profile, the overall response rate for patients dosed intravenously between 20 and 180 ug/kg is currently 78%, and the response rate is 68% for patients dosed subcutaneously between 135 and 405 ug/kg.

JNJ-64007957 (Teclistamab) is currently being examined in six clinical trials, of which results have been published for one. NCT03145181 is examining the effect of Teclistamab as a single agent in patients with relapse refractory MM. Patients who received doses of 270 ug/kg and 720 ug/kg IV and 720 µg/kg and 1500 µg/kg SC on a weekly basis had an ORR of 63.8%, including 9 complete responses (CRs) and 24 very good partial responses or better [124].

## 4. CAR-T Cells

Chimeric Antigen Receptor (CAR) T cells have shown unprecedented response rates in hematological cancers. Since 2017, several CAR-T therapies have been approved by the FDA for pediatric leukemia and adult lymphomas and, just this year, the first CAR-T therapy against multiple myeloma was approved. In the current paradigm, the patient’s own T cells are virally modified to introduce a chimeric receptor directed towards surface molecules of the target tumor cells. This use of an autologous therapy results in reduced concerns regarding rejection of the therapy and a greater potential for prolonged protection. Recent efforts are also focusing on allogenic CAR-Ts and gene-edited CAR-Ts.

### 4.1. Mechanism of Action

The engineered introduction of the CAR molecules into the T cells permits the targeting of tumor cells by pairing non-MHC-dependent molecular recognition with native T cell signaling. This allows for the same cascade to be triggered as when the T cell receptor (TCR) detects its cognate antigen (Figure 2) [125]. This mechanism results in effective targeted killing through granzyme and perforin secretion, similar to the native way in which T cells kill target cells [126]. Given that the targeting is based on either antibody fragments or catalogued libraries of ligand-binding motifs, there is a wide array of potential targets which can be developed into CAR constructs to treat various cancers [125]. Additionally, CAR-T cells have demonstrated killing through the use of the Fas/FasL pathway wherein FasL (CD95L) on the CAR-T cell binds with its receptor Fas (CD95) and induces activation of caspase 8, which forms a death-inducing signaling complex (DISC) in the targeted cell [126]. There data suggest that this mechanism is primarily used following antigen-positive activation of the CAR-T cells, which aids in the removal of antigen negative cancer. However, the threat of off-target killing increases as this pathway is more exploited by the therapeutic cells [127].

### 4.2. History

The initial case reports and successful clinical trials using CAR-T cells for the treatment of cancer consisted of targeting CD19 in cases of relapsed and refractory B cell malignancies [128]. These trials showed incredibly high rates of cancer remission when patients were given CAR-T therapies [129]. Occasionally, the delivered cell product was undetectable within a few months, which tended to result in disease relapse, given the loss of protection by the therapeutic cells. However, the authors of the study validated that the patient’s cancer cells were still sensitive to the same CAR-T cell-killing, meaning that reinfusion of the product or a more persistent cell product would provide protection and induce remission again [129]. Following these initial clinical trials, two anti-CD19 CAR-T cell products gained FDA approval in 2017: tisagenlecleucel from Novartis Pharmaceuticals following their B2202 trial and axicabtagene ciloleucel from Kite Pharma following their ZUMA-1 trial. These approvals further galvanized researchers and companies to operate their own trials using CAR-T to treat other forms of cancer.

### 4.3. CAR-T in Multiple Myeloma

As discussed above, multiple myeloma is one of several cancers for which a CAR-T product is now FDA-approved, and many other clinical trials are underway. The primary target that has been explored in the context of myeloma is the B-cell, maturation antigen (BCMA), given that it has minimal off-target expression (Figure 2) [130]. Additionally, BCMA is essential for plasma cell proliferation and survival. Therefore, it is unlikely to undergo sufficient mutation to allow the cancer cells to escape from the therapy [9]. However, there are also trials exploring the use of other targets, such as NY-ESO-1, CD-19, and GPRC5D [131,132,133]. NY-ESO-1 is a cancer/testis antigen that is highly specific to the cancer cells and has shown to be a useful CAR-T target, but it does require genetic sequencing of the multiple myeloma in order to validate that it is being expressed in the patient. Conversely, CD-19 is very rarely expressed in plasma cells, but CAR-T trials have validated that it seems to be an effective target in the context of multiple myeloma [132]. GPRC5D is also not canonically expressed in plasma cells but has been detected in multiple myeloma patients independent of BCMA expression, meaning that it could serve as a useful secondary target [133].

In March of 2021, the FDA formally approved the first CAR-T therapy for relapsed/refractory multiple myeloma in idecabtagene vicleucel from Bristol Myers Squibb and bluebird bio, following a trial showing an overall response rate of ~70% in the phase 2 trial. The approval of idecabtagene vicleucel and its CD-19 predecessors tisagenlecleucel and axicabtagene ciloleucel further validate that CAR-T cell products will likely continue to gain approval for various cancer types.

### 4.4. Current Drawbacks to CAR-T

While the short-term response rates of CAR-T therapies tend to be encouraging, there is still a significant risk of post-treatment relapse. The primary concern is that CAR-T treatments for multiple myeloma tend to have problems with persistence, where the T cells do not engraft and therefore do not provide prolonged protection. This is emphasized by the fact that in vivo expansion is typically the most consistent factor, correlating with patient responses across clinical trials [134]. The working hypothesis for this issue is that the manufactured T cells display a proportion of memory subphenotypes that is too small to persist and engraft [135]. Another likely cause of post-therapy remission is exhaustion of the CAR-T cells due to overactivation during expansion or anergy following delivery [136]. This is particularly due to the immunosuppressive microenvironment seen in the case of multiple myeloma but not in other diseases for which CAR-T therapy is approved. Antigen loss, where the relapsed disease does not express the original CAR target, is another source of concern as well [137,138]. This is more commonly seen in the case of CD-19-targeting therapies, but recent publications have shown biallelic deletion of BCMA, which have caused therapy to fail upon administration of a second CAR-T infusion [139]. This outcome currently represents a minority of cases in multiple myeloma, as the majority of patients relapse with BCMA^+^ disease, but is still being considered as more data become available from clinical trials [134]. Finally, cancer dysregulation of death receptor signaling, such as FADD, BID, and TRAIL2, has been shown to play a role in CAR-T cell off-target killing. However, its effects in the context of multiple myeloma are still under investigation [127].

There are two key clinical toxicities that have been noted with the delivery of CAR-Ts: CRS and neurotoxicity [140]. While the clinical symptoms of the toxicities vary and can range from mild to life-threatening, the prevailing hypothesis is that both are caused by the same phenomenon. The rapid expansion and activation of the delivered CAR-T cells results in a sudden increase in immunostimulatory cytokines, such as TNF, IFN-γ, and GM-CSF, which activate innate immune cells and trigger a positive feedback loop of these same cytokines, called a cytokine storm [140]. The likelihood and grade of these toxicities is correlated with patient tumor burden, requiring close monitoring of these high-risk patients following treatment administration [128,140]. The clinical management of these toxicities is commonly carried out through the use of immunosuppressive agents such as corticosteroids, cytokine receptor antagonists (anakinra), and antibodies targeting either the cytokines themselves (lenzilumab) or their receptor (tocilizumab) [140].

The bone marrow niche where multiple myeloma proliferates and differentiates is a complex interaction of cells, cytokines, and a noncellular compartment [141]. These interactions actually help the survival of multiple myeloma cells and lead to difficulties with treatment. In the context of CAR-T therapy, the cytokines present in the TME prevent effective CAR-T cell infiltration and alter their ability to locate the desired target cells. These issues are further compounded by the short-term persistence of T cells following delivery to the patient, which limits the durability of the therapy and requires complete eradication of the target in a brief timespan to avoid remission.

The concerns regarding short-term persistence begin with the identification and optimization of the T cells’ sub-phenotypes, which are delivered to the patient following ex vivo expansion. Post-treatment phenotype analyses of cells delivered to responders vs. non-responders have highlighted that patient outcomes tend to be more favorable when there are higher percentages of memory (CD62L^+^ CCR7^+^) cells and a more equitable distribution of CD4 and CD8 cells [142,143]. However, most standard expansion systems used in CAR T cell manufacturing tend to simply target CD3 and CD28 in non-physiologically relevant manners, which tends to disproportionately expand CD8 effector cells [144]. While novel expansion systems are being developed with the goal of improving the phenotypic distribution of the cell product, their implementation in approved therapies would require new trials to formally validate their safety and efficacy [145,146].

Another consideration that affects the quality of the downstream product is the fact that current approval for CAR-T is restricted to cases of relapsed or refractory disease, which results in reduced cell fitness of the starting material and, therefore, a reduced fitness of the final cell product [132]. The advanced disease state of patients eligible for the therapy also causes problems with current manufacturing timelines, where vein-to-vein times could be as long as 4 weeks, depending on the product and the necessary quality control [128]. Further concerns involving the manufacturing of CAR-T cells regard the use of retroviral/lentiviral gene delivery for CAR expression, which can result in variable surface expression or issues with the randomness of the genetic insertion [147]. Additionally, vector fabrication tends to be a time-intensive and expensive process, particularly when considering the need to use larger scales as more therapies are approved [148]. There is some work on the development of mRNA-transduced CAR-T cells that lose expression of the receptor within 14 days to limit adverse effects but this remains in pre-clinical stages and requires extensive validation as a large-scale and durable alternative to viral transduction [149]. Finally, the cost to patients and insurers is a huge hurdle affecting the adoption of CAR-T therapy, given that the average total costs for administration easily exceed $400,000 and can climb as high as $1,000,000 [150].

### 4.5. Future

Researchers are cognizant of the current limitations of CAR-T and plenty of work is being carried out to improve the efficacy and quality of the therapy while expanding its accessibility. One approach to address the threat of antigen escape is to develop dual targeting, such as BCMA and transmembrane activator and CAML interactor (TACI), in the context of multiple myeloma using dual-binding ligands (APRILs), which are then targeted by the CAR [151]. Similarly, the co-expression of anti-BCMA and anti-GPRC5D CARs on the same cell has shown promising results in pre-clinical models of disease relapse [152]. Additionally, logic circuits can be introduced into CAR-T cells using synNotch receptors, which trigger transcriptional expression of the CAR molecule itself upon detection of an initial target [153]. These approaches allow for a more robust detection of multiple targeted surface markers, which increase the specificity of the therapy and should decrease the likelihood of exhaustion post-delivery. Other approaches to addressing exhaustion are the combinatorial delivery of CAR-T cells, along with immune checkpoint blockade or even the co-expression of CAR and IL-7 receptor [154,155].

Additionally, there is a push towards the development of techniques to remove HLA and TCR hurdles, which would cause graft-versus-host-disease (GvHD) in cases where non-donor matched CAR-T cells are delivered [156]. Developing allogeneic CAR-T cells that could be delivered to a wider array of patients would help to revolutionize the implementation of therapy by dramatically reducing its cost, due to the ability to scale-up, and reduce vein to vein times by having the therapy ready off-the-shelf. Finally, there is increased collaboration between industry and academia in order to proactively address manufacturing concerns. One of the largest and most visible collaborations is the NSF-funded Engineering Research Center for Cell Manufacturing Technologies (CMaT), whose work is focused on developing tools, assays, and supply chain solutions to improve the manufacture of cell therapeutics by working with industry and regulatory bodies to ensure facilitated transfer. These kinds of public–private partnerships will aid the alleviation of the financial burden of technology development for industry members—which, in turn, is relayed to patients—while still developing new methods to make these cutting-edge therapies more effective, durable, and cost-effective for the patients.

## 5. Vaccines

The use of vaccines to treat cancer is one of the first forays into cancer immunotherapy [157]. While the strategies to accomplish this vary, the overarching goal is the engagement of the adaptive immune system to eliminate transformed cells. One of the main methods through which this is accomplished is enhancing the presentation of tumor-associated antigens by dendritic cells to T cells [158]. Tumor vaccines in multiple myeloma can take a variety of forms, ranging from idiotype-targeting vaccines, myeloma cell, dendritic cells, fusion vaccines, and peptide-based vaccines. Additionally, multiple strategies are used to determine target antigens. Autologous vaccines utilize a resected tumor from a patient to develop a personalized vaccine that encompasses neo-antigens that are unique to the individual patient. While this strategy provides the best coverage of target antigens, it is impeded by the limited supply of tumor tissue and the need to generate a unique vaccine for each patient [159]. Allogenic vaccines utilize human multiple myeloma cell lines to generate an off-the-shelf product that targets the multiple myeloma antigens that are shared by the majority of patients [159]. Peptide-based vaccines consist of injections of a tumor-associated antigen along with an adjuvant that serves to activate the immune system. This causes antigen-presenting cells to take up the tumor-associated antigen and present it to T cells in the draining lymph node to induce target immunity [160].

### 5.1. Mechanism of Action

The goal of a tumor vaccine is to engage the immune system to target transformed cells. While the mechanisms through which this are accomplished vary, the desired result is directed T cell activation and the establishment of long-term immune memory [161]. The antigens delivered by the vaccine are transported to the draining lymph node by antigen-presenting cells, where T cells sample the presented antigen, inducing the activation of target antigen-specific T cells. This process is enhanced through the use of an immune-stimulating adjuvant such as GM-CSF or IL12 [159].

### 5.2. History

The first demonstration of a tumor vaccine to provide protective immunity occurred in 1978, when Dr. Hanna Jr and Dr. Peters generated immunity in guinea pigs to the L10 hepatocarcinoma cells. They demonstrated that the Bacillus Calmette-Guérin vaccine injected with L10 tumor cells was able to cure test subjects with established metastatic disease [162]. The injection of L1 cells along with Bacillus Calmette-Guérin, however, did not protect against L10-based carcinoma demonstrating the limitations of this approach [162].

Early attempts at developing a vaccine targeting multiple myeloma focused on targeting the unique idiotype expressed by the malignant cells. This approach is attractive as it provides a patient-specific target that can be readily derived from a patient’s serum [163]. Idiotype-pulsed dendritic cells induced protective tumor immunity from myeloma tumor cell lines in animal models; however, this protection did not effectively translate to activity in humans [163]. A clinical trial examining the effect of the idiotype DC-based vaccine APC8020 demonstrated no benefit in terms of progression-free survival [164].

### 5.3. Current Strategies

#### 5.3.1. Dendritic Cell Vaccines

Dendritic cells are the premier antigen presenting cells in the body, and dendritic cell vaccines aim to utilize this to induce cancer immunity [165]. Typically, monocytes from a patient’s peripheral blood are harvested and differentiated into immature dendritic cells, which are then loaded with the tumor-associated antigen and reintroduced to the patient as mature and activated dendritic cells [166]. Dendritic cell vaccines can be loaded with specific tumor-associated antigens to induce targeted immunity, allogenic cell lines that provide multifaceted protection across a variety of antigens, or autologous patient-derived tumor cells, which provide multifaceted protection including patient-specific neoantigens [165,166].

Dendritic cell fusion vaccines that utilize autologous tumor cells to generate a vaccine that captures a tumor’s entire mutanome have shown promise in the post-stem-cell transplant setting. In one study, after receiving an autologous dendritic fusion vaccine post-ASCT, 78% of patients achieved a CR or VGPR, with 47% being CR. A total of 24% of patients who achieved a partial response were converted to CR in the months after receiving the vaccine [167]. A second study in the post-ASCT setting, examining the use of autologous DC fusion vaccines in combination with PD1 inhibition, saw the percentage of CD8+ tumor reactive T cells increase from 1.8% to 9.16% after the administration of immune therapy [168].

#### 5.3.2. Peptide-Based Vaccines

Peptide-based vaccines involve the injection of one or multiple tumor-associated antigens, along with an immune-stimulating adjuvant to induce the processing of the tumor-associated antigen and the subsequent induction of a tumor-specific T cell response. These are examined in the context of both SMM and post-ASCT as a method to halt disease progression and enhance the duration of patient responses, respectively.

PVX-410 utilizes four peptide antigens targeting X-Box Binding protein 1, CD138, and SLAMF7, each of which were demonstrated to induce a myeloma-specific T cell response [169,170]. NCT01718899 examined the use of PVX-410 as a monotherapy and in combination with lenalidomide to treat patients with high-risk SMM. As a monotherapy and in combination, PVX-410 induced a peptide-specific immune response in 91% and 100% of patients, respectively [170]. Patients receiving the PVX-410 monotherapy experienced a median time to progression of 36 weeks, a benchmark that has not yet been reached in the combination group [170]. PVX-410 is further being examined in combination with lenalidomide and the histone deacetylase inhibitor Citarinostat as part of clinical trial NCT02886065, with results yet to be published [171].

The GVAX platform works to combine the concept of a peptide vaccine with the multifaceted coverage of an allogenic vaccine. In this platform, commercially available multiple myeloma cell lines H929 and U266 are injected along with the K562-GM-CSF cell line, which is engineered to overexpress GM-CSF [172]. NCT01349569 examines the GVAX platform in combination with a lenalidomide-containing regimen for use in patients with nCR and stable disease [173]. Patients receiving GVAX demonstrated improved progression-free survival, with median survival not being reached, compared to 10 months in the observation arm, receiving a lenalidomide-containing regimen [172,173]. Additional clinical trials are examining the combination of lenalidomide and the GVAX platform; however, the results are yet to be published.

Additionally, phase 1 clinical trials examining the effects of BCL-2-family-targeted vaccines and PD-L1-targeted vaccines have demonstrated an acceptable safety profile as well as the ability to induce an immune response [174,175].

### 5.4. Future

mRNA vaccines have recently gained fame regarding the prevention of SARS-CoV-2 but also offer a unique opportunity in the treatment of cancer [176,177,178]. Through the use of high-throughput screening, mRNA vaccines can provide a patient-specific vaccine that targets neoantigens unique to each patient’s disease [178]. Patient-specific mRNA can then be delivered through one of two means: direct injection of mRNA or through autologous DCs that have been loaded with disease-specific mRNA. The direct injection of mRNA consists of naked or carrier-linked mRNA being injected into immune active tissue such as a lymph node, either intradermally or intranasally. The mRNA in these immune active sites is then taken up by professional APCs, which then translate the mRNA into protein and, through MHC presentation, induce a tumor-specific immune response. The immune response to naked mRNA is often enhanced through the use of adjuvants such as TriMix, which consists of mRNAs coding for CD70, CD401, and a constitutively active TRL4 [178]. The use of TriMix was associated with enhanced DC maturation and improved CTL response [179]. Similar to the peptide-based vaccines described above, mRNA can also be directly loaded into autologous dendritic cells, which are then reinfused into the patient, where the DCs induce a tumor-specific immune response. This strategy can be enhanced through the use of mRNA-encoding costimulatory molecules such as CD83 and the 4-1BB ligand [178]. At present, in the context of multiple myeloma, NCT01995708 is examining the use of Langerhans-type Dendritic Cells loaded with mRNA encoding for three tumor-associated antigens, CT7, MAGE-A3, and WT1 [165]. Results have not yet been reported.

## 6. Discussion

The past three decades have been a time of discovery and progress in regard to advances in the treatment of patients with multiple myeloma. Immune therapies have delivered particularly incredible patient outcomes in both newly diagnosed and relapsed refractory multiple myeloma. IMiD-based therapy has revolutionized myeloma care and become a crucial component of therapy regimens in transplant-eligible and -ineligible patients alike. Monoclonal antibody-based therapies have capitalized on the highly differentiated state of myeloma cells to engage both the innate and adaptive immune systems to induce disease remission. Antibody drug conjugates also capitalize on the abundance of tumor-associated antigens and allow for the delivery of highly potent and otherwise unusable chemotherapeutic agents. CAR-T and bi-specific antibody therapy have induced clinical responses in heavily pretreated relapse refractory myeloma patients that lack therapeutic options. In SMM and the adjuvant post-stem-cell-transplant setting, vaccines have shown promise in the prevention of disease progression and extended disease remission. Immune therapy is a broad term and, in the context of multiple myeloma, encapsulates a diverse array of therapies that each offer their own therapeutic advantages and challenges. As research continues and a deeper understanding of the immune system and immune microenvironment is achieved, there will continue to be advances that improve the lives of patients with multiple myeloma.

## 7. Conclusions

While advances in therapy have dramatically improved patient outcomes in multiple myeloma over the past 30 years, there is still a tremendous unmet need. Modern immune therapy has utilized a variety of immune agents to increase the gains further and achieve a stronger and more durable patient response. As research continues to improve our understanding of the immune system and hone the tools available, immune therapy will continue to be on the cutting edge of multiple myeloma therapy. IMiDs, monoclonal antibodies, bi-specific antibodies, CAR-T therapy, and cancer vaccines all have an impact on the therapeutic landscape of multiple myeloma and will continue to shape patient outcomes in the future.

## Figures and Tables

**Figure 1 cancers-13-04787-f001:**
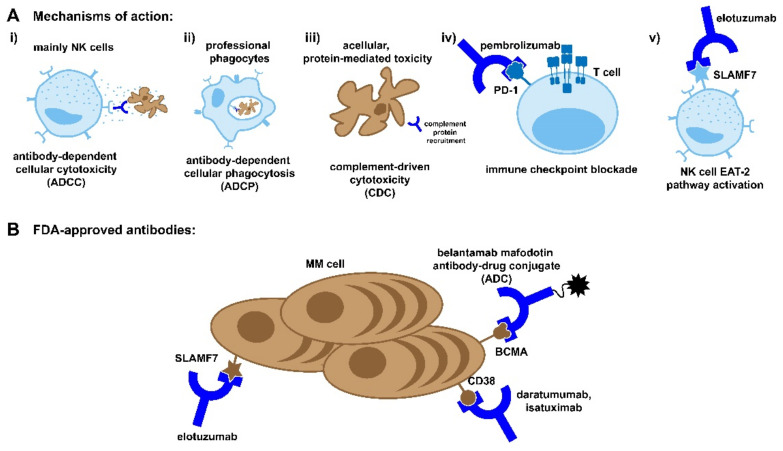
Antibody based therapies. (**A**) The mechanisms of action for monoclonal antibody and antibody-drug-conjugate therapies. As antibodies already play a role in the immune response against pathogens, these therapies harness the same pathways to elicit both cellular- and acellular-mediated toxicity of the target cell. These pathways include canonical killing of the target cell by immune cells ((**i**) ADCC, (**ii**) ADCP, and (**iii**) complement) as well as the “reprogramming” of immune cells to reactivate against the target cell ((**iv**) immune checkpoint blockade and (**v**) EAT-2 activation). (**B**) In addition to checkpoint blockade antibodies, there are several FDA-approved monoclonal antibodies and drug conjugates currently available to treat MM, which target several common markers of myeloma cells.

**Figure 2 cancers-13-04787-f002:**
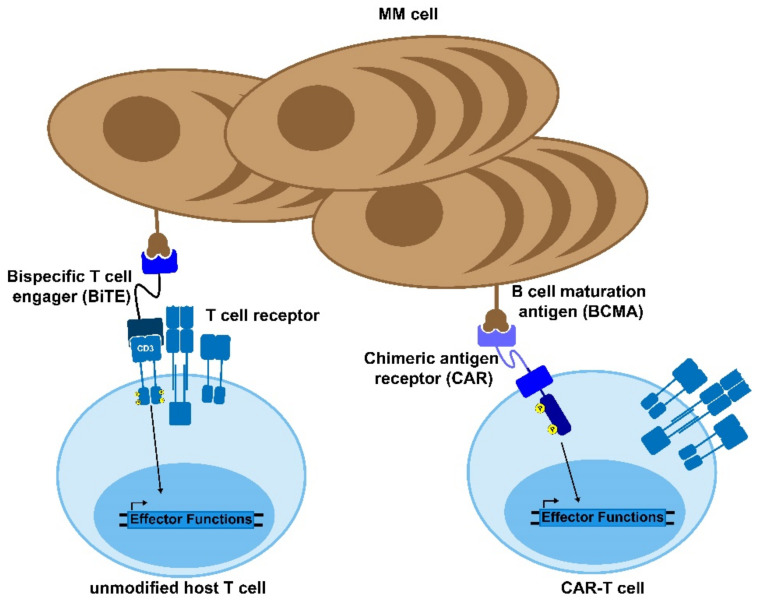
Bi-specific antibodies and CAR-T cells. Bispecific T cell engagers (BiTEs) and chimeric antigen receptor (CAR) T cells are novel therapies to treat MM and other diseases. The BiTE uses two antibody domains to crosslink the target cell to an immune effector cell in order to activate the host immune response against the tumor. Similarly, CAR-T cells are genetically modified to express a chimeric, MM-recognizing antigen that can bypass the normal T cell activation cascade to enable the CAR-T cell to respond to and kill target antigen-expressing cells.

**Table 1 cancers-13-04787-t001:** Current clinical trials examining the use of bi-specific antibodies in multiple myeloma.

Trial Number	Drug	Tumor Target	T Cell Target	Status
NCT03933735	TNB-383B	BCMA	CD3	Recruiting
NCT03269136	PF-06863135	BCMA	CD3	Active, not recruiting
NCT04649359	PF-06863135	BCMA	CD3	Recruiting
NCT04798586	PF-06863135	BCMA	CD3	Recruiting
NCT03309111	ISB 1342	CD38	CD3	Recruiting
NCT04773522	Talquetamab	GPRC5D	CD3	Recruiting
NCT03761108	REGN5458	BCMA	CD3	Recruiting
NCT04083534	REGN5459	BCMA	CD3	Recruiting
NCT04634552	Talquetamab	GPRC5D	CD3	Recruiting
NCT04722146	Teclistamab	BCMA	CD3	Recruiting
NCT04557098	Teclistamab	BCMA	CD3	Recruiting
NCT03399799	Talquetamab	GPRC5D	CD3	Recruiting
NCT04696809	Teclistamab	BCMA	CD3	Recruiting
NCT03145181	Talquetamab	GPRC5D	CD3	Recruiting
NCT04735575	EMB-06	BCMA	CD3	Not yet recruiting
NCT04586426	Talquetamab/ Teclistamab	BCMA/GPRC5D	CD3	Recruiting
NCT04108195	Talquetamab/ Teclistamab	BCMA/GPRC5D	CD3	Recruiting
NCT04910568	RG6160	FcRH5	CD3	Not yet recruiting
NCT03275103	RG6160	FcRH5	CD3	Recruiting
NCT03486067	CC-93269	BCMA	CD3	Recruiting
NCT03287908	AMG 701	BCMA	CD3	Recruiting

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
