# Peer review of "Keeping Myeloma in Check: The Past, Present and Future of Immunotherapy in Multiple Myeloma"

_cancers, 2021, doi:10.3390/cancers13194787_

Round 1

Reviewer 1 Report

Authors wrote a very good review about immunotherapies not only that are currently used to treat multiple myeloma patients but also future therapeutics in the preclinical and clinical trial stage.

Very minor points to make this review better:

Abbreviations such as ASCT, CRBN, and etc, are used but are not written fully when first time they are introduced.

Line 87, “CRBN complexes with CUL4A or CUL4B, DDB1, 87 and Roc1.” Explaining these may help readers to understand better.

Figure 1. There are 5 figures in panel A and figure legend can be more specific and explain details in panel A.

Figure 1 and 2, Red multiple myeloma doesn’t look professional and too big compare too other parts. Modifying them will definitely improve the quality of the figures.

Author Response

Authors wrote a very good review about immunotherapies not only that are currently used to treat multiple myeloma patients but also future therapeutics in the preclinical and clinical trial stage.

We thank the reviewer for their positive review of manuscript and for their constructive feedback.  We have addressed each point below.

Very minor points to make this review better:

Abbreviations such as ASCT, CRBN, and etc, are used but are not written fully when first time they are introduced. 

We have corrected this throughout the manuscript.

Line 87, “CRBN complexes with CUL4A or CUL4B, DDB1, 87 and Roc1.” Explaining these may help readers to understand better.

We have included a description of the components of this culling ring ligases complex.  This can be found on lines 89-96.

Figure 1. There are 5 figures in panel A and figure legend can be more specific and explain details in panel A.

A more detailed figure legend is now provided for Figure 1

Figure 1 and 2, Red multiple myeloma doesn’t look professional and too big compare too other parts. Modifying them will definitely improve the quality of the figures.

The myeloma cell color has been changed and the cells made smaller.

Reviewer 2 Report

Simple summary:

page 1, line 18: grammar: “relapse” should be “relapsed”

Abstract:

Reads well, no issues.

Introduction:

Page 2, line 53: grammar: remove “many” from the sentence.

Page 2, line 56: grammar: “relapse” should be “relapsed”

Otherwise reads well.

IMIDs:

Page 3, line 76: grammar: remove “are” from the sentence.

In the beginning section after the discussion of T-cell activation, you may want to mention that the T cell activation has been shown to induce GVHD when used post-allo as an example of a potential downside to the increased T cell activity.

Thalidomide:

1st sentence.  Thalidomide also has FDA-approved upfront treatment indications as well.

Would also suggest mentioning that thalidomide is the only IMiD not associated with significant marrow suppression, thus making it a good choice for Jehovah’s witnesses or others that cannot accept blood product support.  You mention poor tolerability of thalidomide in the len section, but don’t explain what the thal tolerability issues are.  Would mention fatigue, brain fog, constipation, irreversible peripheral neuropathy in the thal section to flesh out.

Pom:

True that metabolism in the liver increases risk of drug interactions, but that gives pom the advantage of being easier to dose in those patients with renal insufficiency, whereas len would require dose reduction.  You may want to include this information in this section.

SMM:

  • Line 186. The staging system was published a collaboration within the IMWG, not exclusively Mayo clinic: Blood Cancer J. 2020 Oct 16;10(10):102.
  • You may want to mention the risk system for SMM incorporating cytogenetics (1q) in addition to the 2/20/20.
  • Any overall survival data you can provide for early treatment of SMM?

Elotuzumab:

  • Would include the data from the ELOQUENT-3 study (elo + pom, NEJM 2018) which was a very powerful combination to use in RRMM. This is also an FDA approved regimen.

Daratumumab:

  • There is enough data from the upfront Griffin study to include here.

Isatuximab:

  • This section is anemic. You should include data from the recent IKEMA and ICARIA studies.

Would also mention the other difficulties with anti-CD38 therapy: complications with blood typing and also interpretation of SPEP with a therapeutic band present.

Belantamab:

  • Would mention data from DREAM-6 study showing enhanced activity in combination with bortezomib.

Document flow: Swap the MEDI2228 and “Future” paragraphs in the ADC section.

BiSpecific Antibodies:

  • You write: AMG-420 continues to be evaluated with alternative dosing regimens. I believe that this is not the case and that AMG-420 is not being pursued anymore by Amgen.  Existing trials are not recruiting and no new trials are planned.
  • Page 13, line 560: typo: “unprecdented”

BiSpecifics and Car-Ts:

  • The major immediate toxicities of cytokine release syndrome and neurotoxicity are not mentioned in these sections. A discussion is needed, along with management with agents such as tocilizumab. The issues with use of corticosteroids to manage toxicities also deserve mention.

Vaccines:

  • No issues, reads well.

Discussion:

  • No issues, reads well.

Author Response

We thank the reviewer for the thorough review of the manuscript and for the constructive comments.  We have addressed each below. Please note, that the line numbers represent those when the manuscript is in full markup display.  Unfortunately the numbers change when the markup display

Simple summary:

page 1, line 18: grammar: “relapse” should be “relapsed”

This has been corrected (line 21)

Abstract:

Reads well, no issues.

 Thank you

Introduction:

Page 2, line 53: grammar: remove “many” from the sentence.

This has been corrected (line 55)

Page 2, line 56: grammar: “relapse” should be “relapsed”

This has been corrected (line 58)

Otherwise reads well.

 Thank you

IMIDs:

Page 3, line 76: grammar: remove “are” from the sentence.

This was removed (line 78)

In the beginning section after the discussion of T-cell activation, you may want to mention that the T cell activation has been shown to induce GVHD when used post-allo as an example of a potential downside to the increased T cell activity.

We considered this however decided not to include a discussion of GVHD since allogeneic transplant is no longer routinely used in myeloma treatment. At this point it is only used as part of clinical trials.

Thalidomide:

1st sentence.  Thalidomide also has FDA-approved upfront treatment indications as well.

We have edited the sentence (lines 134-135)

Would also suggest mentioning that thalidomide is the only IMiD not associated with significant marrow suppression, thus making it a good choice for Jehovah’s witnesses or others that cannot accept blood product support.

We thank the reviewer for the comment and have included a statement on lines 148-150

  You mention poor tolerability of thalidomide in the len section, but don’t explain what the thal tolerability issues are.  Would mention fatigue, brain fog, constipation, irreversible peripheral neuropathy in the thal section to flesh out.

This is now included on lines 137-140

Pom:

True that metabolism in the liver increases risk of drug interactions, but that gives pom the advantage of being easier to dose in those patients with renal insufficiency, whereas len would require dose reduction.  You may want to include this information in this section.

This is now included on lines 187-189.

SMM:

  • Line 186. The staging system was published a collaboration within the IMWG, not exclusively Mayo clinic: Blood Cancer J. 2020 Oct 16;10(10):102.

We thank the reviewer for pointing this out and it is now included on line 214 and reference 44.

  • You may want to mention the risk system for SMM incorporating cytogenetics (1q) in addition to the 2/20/20.

This is now included on lines 221-225

  • Any overall survival data you can provide for early treatment of SMM?

We have included OS data on lines 235-237 from reference 45

“While median overall survival was not reached, at the time of data cut off 18% of patients died in the treatment group compared to 36% in the control group”

Elotuzumab:

  • Would include the data from the ELOQUENT-3 study (elo + pom, NEJM 2018) which was a very powerful combination to use in RRMM. This is also an FDA approved regimen.

We thank the reviewer for the suggestion.  Discussion of ELOQUENT-3 is now included as a paragraph at the end of the section (lines 313-322)

Daratumumab:

  • There is enough data from the upfront Griffin study to include here.

Daratumumab is also being evaluated in combination with lenalidomide, bortezomib, and dexamethasone for use in induction, consolidation, and maintenance therapy in transplant eligible NDMM as part of the GRIFFIN trial (NCT02874742).  42.2% of patients who received the daratumumab achieved a sCR compared to 32% in the control arm.  Daratumumab also increased the overall response rate from 91.8% to 99%. 

Thank you for the suggestion, this is now included as a paragraph in this section (lines 355-361)

Isatuximab:

This section is anemic. You should include data from the recent IKEMA and ICARIA studies.

We have significantly expanded this section including data from IKEMA and ICARIA (lines 372-383).

Would also mention the other difficulties with anti-CD38 therapy: complications with blood typing and also interpretation of SPEP with a therapeutic band present.

We have included a section on blood typing (lines 379-383).  We also discuss the issues with SPEP and the use of mass spectrometry to overcome this (line 257-260)

Belantamab:

  • Would mention data from DREAM-6 study showing enhanced activity in combination with bortezomib.

DREAM-6 data is now included (lines 485-488)

Document flow: Swap the MEDI2228 and “Future” paragraphs in the ADC section.

We made the change as suggested.

BiSpecific Antibodies:

  • You write: AMG-420 continues to be evaluated with alternative dosing regimens. I believe that this is not the case and that AMG-420 is not being pursued anymore by Amgen.  Existing trials are not recruiting and no new trials are planned.

We have updated the statement to say that while the trial is active it is no longer recruiting and development has been halted (lines 700-701)

  • Page 13, line 560: typo: “unprecdented”

This is corrected

BiSpecifics and Car-Ts:

  • The major immediate toxicities of cytokine release syndrome and neurotoxicity are not mentioned in these sections. A discussion is needed, along with management with agents such as tocilizumab. The issues with use of corticosteroids to manage toxicities also deserve mention.

This is now discussed in both the sections (lines  662-673 and 827-840)

Vaccines:

  • No issues, reads well.

Thank you

Discussion:

  • No issues, reads well.
  •  

Thank you

Added Conclusions as was requested on the document

This section was added after the Discussion (lines 1038-1047)